# MEDVISION: DATASET AND BENCHMARK FOR QUANTITATIVE MEDICAL IMAGE ANALYSIS

## ABSTRACT

Current vision-language models (VLMs) in medicine are primarily designed for categorical question answering (e.g., "Is this normal or abnormal?") or qualitative tasks (e.g., "Describe the image"). However, clinical decision-making often relies on quantitative assessments, such as measuring the size of a tumor or the angle of a joint, from which physicians draw their own diagnostic conclusions. This quantitative reasoning capability remains underexplored and poorly supported in existing VLMs. In this work, we introduce MedVision, a large-scale dataset and benchmark specifically designed to evaluate and improve VLMs on quantitative medical image analysis. MedVision spans 22 public datasets covering diverse anatomies and modalities, with **30.8 million** image-annotation pairs. We focus on three representative quantitative tasks: (1) detection of anatomical structures and abnormalities, (2) tumor/lesion (T/L) size estimation, and (3) angle/distance (A/D) measurement. Our benchmarks show that current off-the-shelf VLMs perform poorly on these tasks. However, with supervised fine-tuning on MedVision, we significantly enhance their performance across detection, T/L estimation, and A/D measurement, demonstrating reduced error rates and improved precision. This work provides a foundation for developing VLMs with robust quantitative reasoning capabilities in medical imaging.

## 1 INTRODUCTION

Recent advances in vision-language models (VLMs) (Li et al., 2024a; Liu et al., 2023; Bai et al., 2023; Zong et al., 2024) have made it possible to pair powerful visual encoders with flexible natural language reasoning, enabling a range of clinical applications such as image-level classification and interactive question answering. In practice, however, most medical VLM usage and evaluation has concentrated on categorical (Lau et al., 2018; Liu et al., 2021; Ben Abacha et al., 2019) or qualitative outputs (Subramanian et al., 2020): answering whether an image is "normal or abnormal," labeling a finding, or producing free-text descriptions of image content. These tasks are important, but they do not capture a critical part of routine clinical reasoning: the extraction of precise, *quantitative* measurements from images that clinicians use to stage disease, plan interventions, and monitor therapy.

Quantitative assessment is central to many medical decisions. Examples include estimating tumor diameter to determine cancer stage, measuring joint angles for orthopedic planning, and computing lesion volume. Clinicians seldom make a diagnosis based solely on a binary label; instead they combine measured quantities with clinical context to reach a conclusion. Despite this, the ability of modern VLMs to produce reliable, reproducible quantitative measurements (e.g., lengths, areas, volumes, angles) has received little systematic attention. Existing benchmarks (Wu et al., 2025; Zhou et al., 2025) and datasets (Lau et al., 2018; Ben Abacha et al., 2019; 2021; Liu et al., 2021; He et al., 2020a; Irvin et al., 2019; Chambon et al., 2024; Johnson et al., 2019; Kayser et al., 2022; Boecking et al., 2022; Lin et al., 2023) emphasize classification accuracy or natural language quality rather than measurement precision, and off-the-shelf VLMs are rarely evaluated on tasks that require exact spatial localization and numeric regression.

There are several reasons why quantitative vision–language capability is both challenging and under-explored. First, measurement tasks demand precise localization (often at sub-pixel or small-structure scales) and robust handling of modality-specific image characteristics (CT, MRI, X-ray, ultrasound).

Second, producing correct measurements requires the model to understand geometric relationships and physical units, not just semantic categories. Third, public datasets with standardized, high-quality measurement annotations are sparse or fragmented across modalities and anatomies, making large-scale evaluation and supervised fine-tuning difficult. Finally, many VLM architectures and training objectives are not designed to produce calibrated numeric outputs or to combine localized visual evidence with explicit numeric reasoning.

To address these gaps, we introduce **MedVision**, a large-scale dataset and benchmark specifically created to assess and improve VLM performance on quantitative medical-image tasks. MedVision aggregates *22* public datasets spanning multiple anatomies and imaging modalities (e.g., radiography, CT, MRI, ultrasound), comprising **30.8 million** images with rich, structured annotations suitable for measurement tasks. We organize the benchmark around three representative, clinically relevant tasks: (1) detection of anatomical structures and abnormalities (localization and identification), (2) tumor/lesion (T/L) size estimation (bidirectional dimensions), and (3) angle/distance (A/D) measurement (e.g., joint angles, inter-structure distances). These tasks capture the range of quantitative demands found in routine radiology and specialty workflows.

Using MedVision, we perform a systematic evaluation of both open-weight and commercial VLMs in two settings: off-the-shelf inference and supervised fine-tuning on the MedVision training splits. Our benchmarks show that off-the-shelf VLMs, even those with strong qualitative language abilities, perform poorly on quantitative tasks, failing at precise localization and producing large numeric errors. We then demonstrate that targeted supervised fine-tuning on MedVision materially improves model performance across detection metrics (recall, precision, F1, IoU) and measurement accuracy (reduced absolute and relative errors for size and angle/distance). Alongside quantitative results, we analyze common failure modes and identify persistent challenges that future research must address.

To summarize, our main contributions are as follows:

- We highlight a clinically important but underappreciated gap: modern VLMs are not reliably able to produce precise quantitative measurements from medical images.

- We introduce MedVision, a large-scale, multi-modality dataset and benchmark for quantitative medical image analysis, covering 22 public datasets and 30.8M images with structured measurement annotations.

- We provide the first comprehensive evaluation of contemporary VLMs on detection, tumor/lesion size estimation, and angle/distance measurement, demonstrating the limitations of off-the-shelf models.

- We release the benchmark (data splits, evaluation code, and baselines) to enable further research toward VLMs that support clinically useful quantitative reasoning.

## 2 MEDVISION: DATASET AND BENCHMARK

An overview of our MedVision dataset and benchmark design is illustrated in Figure 1. We curated a large-scale multi-anatomy and multi-modality medical image dataset with quantitative annotations (Section 2.1). We designed a benchmark to evaluate VLMs' ability on three quantitative medical image analysis tasks (Section 2.2). The benchmark is conducted in the form of open-ended VQA. We convey physical spacing information (i.e., pixel size) in the text prompt to provide VLMs with necessary context for quantitative analysis tasks.

### 2.1 DATASET CONSTRUCTION

We constructed the MedVision dataset with public medical imaging data and annotations. Details on data inclusion criteria, image preprocessing, data annotation, and codebase are described below.

**Inclusion Criteria:** Public medical imaging data with pixel-level annotations (such as mask or landmark coordinate) were collected. We restricted the medical image modalities to those with physical spacing information in the image file header, which is essential for generating ground truth quantitative measurements from images.

We collected 22 datasets, summarized in Table 1, covering a diverse range of anatomies and modalities: AbdomenAtlas (Li et al., 2024b), AbdomenCT-1K (Ma et al., 2022), ACDC (Bernard et al.,

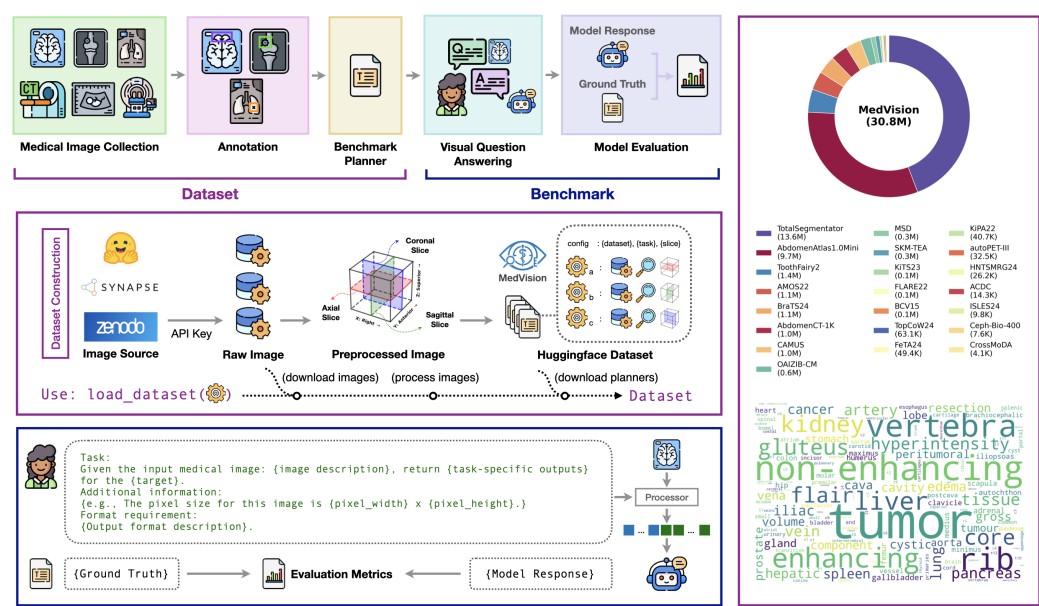

Figure 1: Overview of MedVision dataset construction and benchmark design (left), and dataset summary (right). The MedVision dataset was constructed for out-of-the-box use: `load_dataset` is all you need. Appendix A.1 shows examples of imaging data and quantitative annotation.

2018), AMOS22 (Ji et al., 2022), autoPET-II [1], BCV15 [2], BraTS24 (de Verdier et al., 2024; La-Bella et al., 2025; Kazerooni et al., 2024), CAMUS (Leclerc et al., 2019), Ceph-Bio-400 (Lindner et al., 2016), CrossModDA (Dorent et al., 2023), FeTA24 (Payette et al., 2023), FLARE22 (Ma et al., 2024), HNTSMRG24 [3], ISLES24 [4], KiPA22 (He et al., 2021; 2020b; Shao et al., 2011; 2012), KiTS23 (Heller et al., 2023), MSD (Antonelli et al., 2022), OAIZIB-CM (Ambellan et al., 2019; Yao et al., 2024; Yao & Chen, 2025), SKM-TEA (Desai et al., 2021), ToothFairy2 (Bolelli et al., 2025; 2024; Lumetti et al., 2024), TopCoW24 (Yang et al., 2025), and TotalSegmentator (Wasserthal et al., 2023).

**Image Preprocessing:** We converted the raw imaging data to the standard format and applied standardization as follows. Data are stored in 3D volumes with image orientation corrected to RAS+ convention where the first, second, and third dimensions correspond to the left-to-right, posterior-to-anterior, and inferior-to-superior directions, respectively. Images are neither resampled, cropped, padded, nor rescaled. Our data loading codebase supports flexible image slicing along any of the three anatomical planes (sagittal, coronal, and axial).

**Quantitative Annotations:** We generated bounding box (b-box), bidirectional tumor/lesion (T/L) size, and angle/distance (A/D) measurements for detection, T/L size and A/D estimate tasks, respectively.

*(i) Bounding Box Annotation:* For each segmentation label in a 2D slice, a bounding box is fitted to the mask. Metadata of b-box, including the coordinates of the bottom-left and upper-right corners, pixel and physical dimensions are recorded in the benchmark planner. For labels with scattered clusters, b-box for each cluster is recorded in the released MedVision dataset. However, we excluded 2D slices with multiple b-boxes in our benchmark since we are focusing on single instance detection in this study. Targets with less than 10 pixels in any dimension are excluded from our benchmark.

---

[1] https://autopet-iii.grand-challenge.org
[2] https://doi.org/10.7303/syn3193805
[3] https://hntsmrg24.grand-challenge.org
[4] https://isles-24.grand-challenge.org

*(ii) Tumor/Lesion Size Annotation:* Bidirectional measurements of tumor/lesion from medical images are useful metrics and can provide quantitative information for disease assessment and monitoring. We generated T/L size annotations by calculating the lengths of the major and minor axes of the fitted ellipse to each T/L label in 2D slices. Ellipse fitting is conducted in real-world coordinate system, so that the fitting process takes into account the physical spacing and results in accurate measuring directions (i.e., major and minor axes). Fitted ellipses whose endpoints are outside the buffer zone between the 10% shrunk and enlarged bounding boxes of the target region are excluded to avoid unreliable annotations. The endpoint coordinates of the two axes (in pixel indices) and physical lengths are recorded in the benchmark planner.

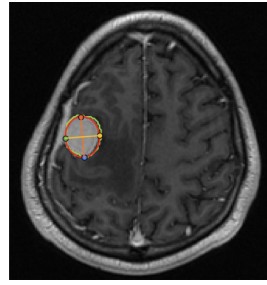

Figure 2: Tumor/lesion size annotation.

*(iii) Angle/Distance Annotation:* Angle and distance measurements are calculated from human annotated landmarks. We collected data from the Ceph-Bio-400 and FeTA24 datasets and generated standardized annotations in the format of landmark coordinates and landmark/angle/distance maps, where each measurement is well-defined and can be calculated from landmark coordinates. Figure 3 shows examples of landmarks in the two datasets. We used the original landmark labels in Ceph-Bio-400 (Lindner et al., 2016). In FeTA24, we assigned labels (P0-P10) to 10 landmarks, where each pair of landmarks defines a distance. Note that each label unambiguously refers to a specific anatomical points, e.g., P0 and P1 denote the most anterior and posterior points of the corpus callosum, respectively.

**Dataset Splits:** MedVision dataset contains **30.8 million** images-annotation pairs across 22 data sources and 3 tasks. For each task, we randomly split the data into training (70%) and test sets (30%) at the patient level. We use the training set for model finetuning where applicable and test set for evaluation.

**Dataset Codebase:** MedVision is built for transparency, version control, easy expansion, and simple use (Figure 1). Its dataset codebase supports reproducible construction and automated loading so users can get started with a single

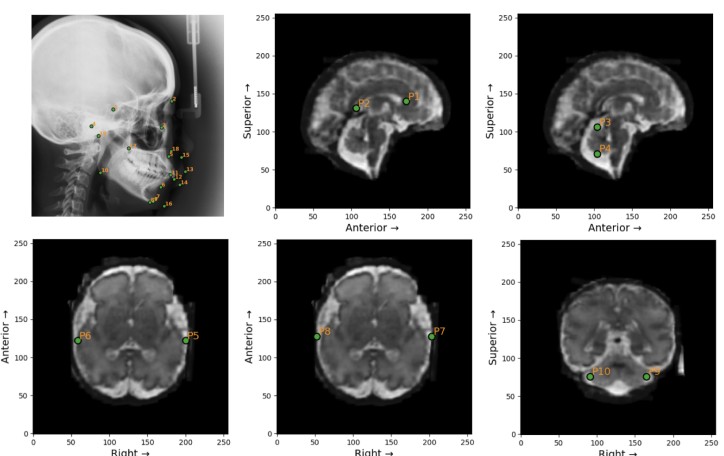

Figure 3: Landmarks in the Ceph-Bio-400 (top-left) and FeTA24 datasets.

call: load_dataset(). A small data configuration (dataset + task + plane) selects the subset to load, while the codebase handles downloading, preprocessing, and preparing benchmark splits.

## 2.2 BENCHMARK DESIGN

We focus on clinically related open-ended VQA tasks for our benchmark: (1) healthy anatomical structures and abnormalities detection, (2) T/L size estimation, and (3) A/D measurement. Model outputs are parsed to extract the required estimates and compared with ground truths.

**Baseline:** We benchmarked 15 state-of-the-art open-weight and commercial VLMs as baselines. Open-weights models include Qwen2.5-VL (Bai et al., 2025), Lingshu (Team et al., 2025b), InternVL3 (Zhu et al., 2025), Gemma3 (Team et al., 2025a), MedGemma (Sellergren et al., 2025), Llama3.2-Vision, LLaVA-OneVision (Li et al., 2024a), LLaVA-Med (Li et al., 2023), MedDr (He et al., 2024), HuatuoGPT-Vision (Chen et al., 2024), and HealthGPT-L14 (Lin et al., 2025). We

Table 1: The MedVision dataset consists of public medical images and quantitative annotations from this study. MRI: Magnetic Resonance Imaging; CT: Computed Tomography; PET: positron emission tomography; US: Ultrasound; b-box: bounding box; T/L: tumor/lesion size; A/D: angle/distance; HF: HuggingFace; GC: Grand-Challenge; † redistributed.

| Dataset | Anatomy | Modality | Annotation | Availability | Source | # Sample (Train / Test) | | |
|---|---|---|---|---|---|---|---|---|
| | | | | | | b-box | T/L | A/D |
| AbdomenAtlas | abdomen | CT | b-box | open | HF | 6.8 / 2.9M | 0 | 0 |
| AbdomenCT-1K | abdomen | CT | b-box | open | Zenodo | 0.7 / 0.3M | 0 | 0 |
| ACDC | heart | MRI | b-box | open | HF†, others | 9.5 / 4.8K | 0 | 0 |
| AMOS22 | abdomen | CT, MRI | b-box | open | Zenodo | 0.8 / 0.3M | 0 | 0 |
| autoPEI-III | whole body | CT, PET | b-box, T/L | open | HF†, others | 22 / 9.7K | 0.5 / 0.2K | 0 |
| BCV15 | abdomen | CT | b-box | open | HF†, Synapse | 71 / 30K | 0 | 0 |
| BraTS24 | brain | MRI | b-box, T/L | open | HF†, Synapse | 0.8 / 0.3M | 7.9 / 3.1K | 0 |
| CAMUS | heart | US | b-box | open | HF†, others | 0.7 / 0.3M | 0 | 0 |
| Ceph-Bio-400 | head and neck | X-ray | b-box, A/D | open | HF†, others | 0 | 0 | 5.3 / 2.3K |
| CrossModDA | brain | MRI | b-box | open | HF†, Zenodo | 3.0 / 1.0K | 0 | 0 |
| FeTA24 | fetal brain | MRI | b-box, A/D | registration | Synapse | 34 / 15K | 0 | 0.2 / 0.1K |
| FLARE22 | abdomen | CT | b-box | open | HF†, others | 72 / 33K | 0 | 0 |
| HNTSMRG24 | head and neck | MRI | b-box, T/L | open | Zenodo | 18 / 6.6K | 1.0 / 0.4K | 0 |
| ISLES24 | brain | MRI | b-box | open | HF†, GC | 7.3 / 2.5K | 0 | 0 |
| KiPA22 | kidney | CT | b-box, T/L | open | HF†, GC | 26 / 11K | 2.1 / 1.0K | 0 |
| KiTS23 | kidney | CT | b-box, T/L | open | HF†, GC | 80 / 35K | 5.9 / 2.6K | 0 |
| MSD | multiple | CT, MRI | b-box, T/L | open | others | 0.2 / 0.1M | 5.3 / 2.2K | 0 |
| OAIZIB-CM | knee | MRI | b-box | open | HF | 0.5 / 0.2M | 0 | 0 |
| SKM-TEA | knee | MRI | b-box | registration | others | 0.2 / 0.1M | 0 | 0 |
| ToothFairy2 | tooth | CT | b-box | registration | others | 1.0 / 0.4M | 0 | 0 |
| TopCoW24 | brain | CT, MRI | b-box | open | HF†, Zenodo | 43 / 20K | 0 | 0 |
| TotalSegmentator | multiple | CT, MRI | b-box | open | HF†, Zenodo | 9.6 / 4.0M | 0 | 0 |
| **Total** | | | | | | **22 / 9.2M** | **23 / 9.6K** | **5.6 / 2.4K** |

evaluated Gemini2.5-Flash under the setting of with and without tool use. Details of these models are provided in Appendix A.2.

**Supervised Finetuning (SFT):** We define task-specific data splits for training and testing. Detection and T/L size tasks use axial slices, while A/D measurement uses all three planes due to limited data. Fine-tuning employed LoRA (Hu et al., 2021) with $r = 16$, $\alpha = 16$, dropout $0.05$, applied to all linear projections, with the embedding layer and LM head trainable.

**Out-of-Distribution (OOD) Performance:** We evaluated the OOD performance of baselines and SFT models on 3 tasks using 2 types of OOD data: (1) **plane-OOD**, where models are evaluated on coronal and sagittal slices after finetuned only on axial slices; (2) **target-OOD**, where models are evaluated on holdout datasets with unseen targets.

**Prompt Design:** We designed text prompts to convey the image description (anatomy and modality, such as "brain MRI"), task instruction (task type and target, such as "bounding box of brain tumor"), format requirement, and optionally physical spacing information. Physical spacing is not provided in detection tasks since the expected output is relative position of b-box in $[0, 1]$. Prompt templates for the detection, T/L size, and A/D measurement tasks are provided in Appendix A.3.

**Physical Spacing Information:** A key feature of our dataset is the inclusion of quantitative annotations, which requires careful handling of physical spacing in text prompts. Since each VLM applies its own image preprocessing, we provide spacing values that match the actual model input. For fixed-size pipelines, adjusted pixel size is derived from input-output dimensions, while for dynamic pipelines, we use the model's image processor to compute the resize ratio. Appendix A.4 details these strategies.

**Evaluation Metrics:** Model performance is evaluated with task-specific metrics. For detection tasks, we report the region-based recall, precision, F1 score, and intersection over union (IoU) for assessment of location accuracy. These metrics are calculated only on the successfully parsed outputs. To reflect the model's ability to follow instructions, we also report the success rate (SR) of generating valid numerical outputs, such as coordinates. We also report $IoU_{>0.5}$, the proportion of samples with IoU greater than 0.5 among all samples. Therefore, $IoU_{>0.5}$ reflects both instruction

Table 2: VLM performance on detection tasks. Targets are grouped into health anatomy and tumor/lesion detection tasks. Mean metrics weighted by sample sizes are reported in %. R: recall; P: precision; F1: F1 score; IoU: intersection over union; SR: success rate.

| Model | Anatomy | | | | | | Tumor/Lesion | | | | | |
|---|---|---|---|---|---|---|---|---|---|---|---|---|
| | R ↑ | P ↑ | F1 ↑ | IoU ↑ | SR ↑ | IoU$_{>0.5}$ ↑ | R ↑ | P ↑ | F1 ↑ | IoU ↑ | SR ↑ | IoU$_{>0.5}$ ↑ |
| Qwen2.5-VL (7B) | 50.4 | 9.5 | 12.4 | 7.6 | 100 | 1.4 | 54.5 | 3.4 | 5.6 | 3.1 | 100 | 0.0 |
| Qwen2.5-VL (32B) | 35.2 | 13.6 | 16.1 | 11.0 | 99.7 | 5.7 | 38.0 | 6.2 | 8.4 | 5.1 | 99.9 | 0.7 |
| Lingshu (32B) | 26.4 | 19.1 | 17.3 | 11.6 | 100 | 4.0 | 23.1 | 5.9 | 7.2 | 4.4 | 100 | 0.6 |
| InternVL3 (38B) | 24.5 | 14.5 | 14.5 | 9.8 | 100 | 5.3 | 39.2 | 5.9 | 7.6 | 4.5 | 100 | 0.4 |
| Gemma3 (27B) | 33.3 | 17.4 | 19.4 | 13.7 | 100 | 9.9 | 33.4 | 5.1 | 6.6 | 4.0 | 100 | 0.5 |
| MedGemma (4B) | 66.2 | 16.3 | 19.0 | 12.8 | 100 | 7.2 | 72.1 | 5.2 | 8.6 | 4.9 | 99.6 | 0.2 |
| MedGemma (27B) | 65.6 | 17.3 | 19.3 | 12.9 | 100 | 6.5 | 65.8 | 5.7 | 8.8 | 5.1 | 100 | 0.2 |
| Llama3.2-Vision (11B) | 47.0 | 8.1 | 10.4 | 6.8 | 73.8 | 2.7 | 45.6 | 2.1 | 3.6 | 2.0 | 70.1 | 0.0 |
| LLava-OneVision (72B) | 36.4 | 17.8 | 18.5 | 12.3 | 100 | 4.4 | 38.1 | 6.0 | 8.5 | 5.1 | 100 | 0.4 |
| LLaVA-Med-v1.5 (7B) | 60.7 | 15.7 | 18.6 | 12.6 | 99.1 | 7.0 | 50.6 | 4.7 | 7.3 | 4.3 | 89.0 | 0.8 |
| MedDr (40B) | 64.4 | 11.9 | 17.3 | 11.3 | 99.8 | 4.1 | 74.9 | 4.4 | 7.5 | 4.3 | 97.8 | 0.1 |
| HuatuoGPT-Vision (34B) | 25.3 | 20.5 | 16.3 | 10.7 | 100 | 3.6 | 20.8 | 5.9 | 7.0 | 4.3 | 100 | 0.6 |
| HealthGPT-L14 | 21.0 | 15.2 | 13.5 | 8.3 | 100 | 0.5 | 22.7 | 6.2 | 6.8 | 4.2 | 100 | 0.5 |
| Gemini2.5-Flash (w/o tool) | 35.4 | 16.1 | 18.7 | 12.8 | 99.5 | 6.8 | 41.4 | 7.1 | 10.1 | 6.3 | 98.3 | 1.2 |
| Gemini2.5-Flash (w tool) | 29.9 | 13.0 | 15.2 | 10.6 | 82.0 | 5.8 | 38.4 | 9.1 | 10.2 | 6.9 | 77.8 | 3.6 |
| Qwen2.5-VL (7B, SFT$_{500K}$) | **79.8** | 76.1 | 76.0 | **68.9** | 100 | 76.4 | 52.7 | 44.2 | 43.3 | 35.0 | 100 | 38.1 |
| Qwen2.5-VL (32B, SFT$_{500K}$) | 76.9 | **79.8** | **76.3** | **68.9** | **100** | **76.9** | 50.8 | **48.2** | **44.7** | **36.6** | **100** | **39.7** |

following and localization ability. For T/L size and A/D measurement tasks, we report the (mean) relative error (M)RE, absolute error (AE), SR and (M)RE$_{<k}$. (M)RE$_{<k}$ refers to the proportion of samples with (M)RE less than $k$ among all samples.

## 3 RESULTS & ANALYSIS

In this section, we are interested in answering the following questions:

- **Q1**: How accurately can VLMs detect *anatomical structures* and *tumors/lesions*?
- **Q2**: Can VLMs estimate the *size* of tumors/lesions by predicting the lengths of their major and minor axes?
- **Q3**: Can VLMs measure *angles and distances* in medical images?
- **Q4**: Can supervised fine-tuning improve the performance of VLMs?
- **Q5**: How do VLMs perform these tasks in out-of-distribution settings?

### 3.1 DETECTION PERFORMANCE

**Baselines vs SFT Models:** Table 2 summarizes detection performance, with targets grouped into healthy anatomical structures and tumors/lesions. Detailed evaluation metrics are provided in Appendix A.5. Most VLMs can produce valid numerical outputs (i.e., bounding-box corner coordinates) with high success rates (SR $> 90\%$), except Llama-3.2-Vision, which achieves only $\sim 70\%$. However, baseline localization accuracy is low, with IoU below $15\%$ for healthy anatomical structures and below $10\%$ for tumors/lesions. After finetuning with 500K samples from our dataset, the SFT models achieved substantially higher recall, precision, F1, IoU, and IoU$_{>0.5}$ compared to baselines (Figure 4).

**OOD Performance:** Figure 5 summarizes the OOD performance of pretrained and finetuned Qwen2.5-VL (32B). The pretrained model serves as a baseline for OOD performance. We observed better generalization ability of the SFT model, indicated by higher precision and F1 score on both plane-OOD and target-OOD data.

**Failure Mode Analysis:** Despite overall improved detection ability of SFT models, they still struggle in some targets. To further understand the failure modes, we analyzed anatomy-level (Appendix A.5.1) and label-level (Appendix A.5.2) performance, and investigated the impact of target size on detection (Appendix A.5.3). We found VLM localization ability increases positively with relative target size and small objects ($< 5\%$ in relative size) detection is the bottleneck in current VLMs.

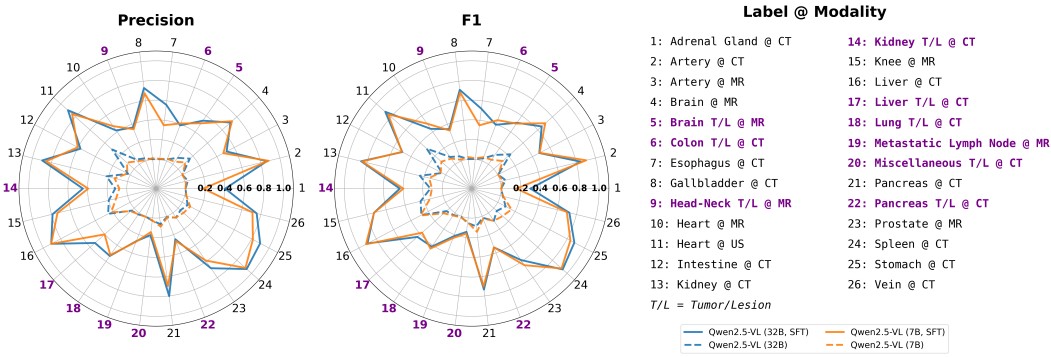

Figure 4: SFT improves VLM performance on healthy anatomical structures and tumor/lesion detection tasks.

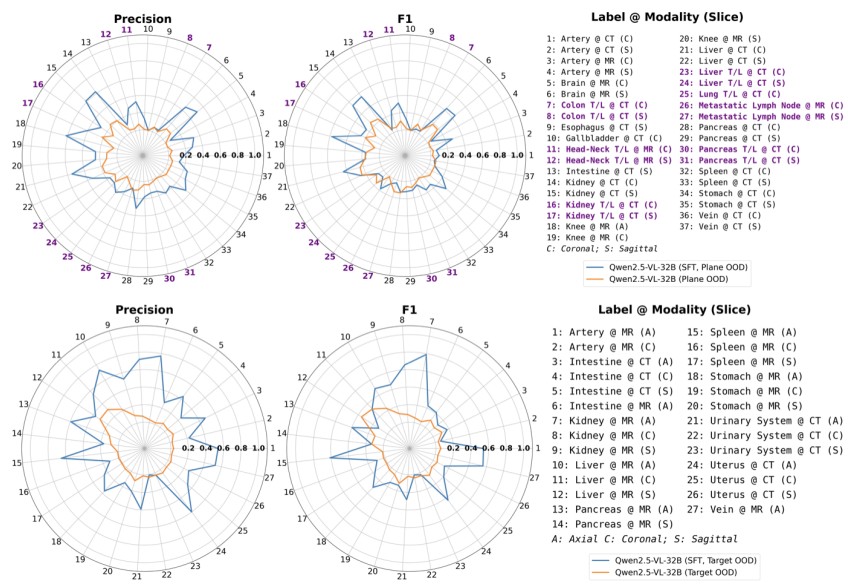

Figure 5: Detection performance on plane-OOD (top) and target-OOD (bottom) data.

**Conclusion:** Pretrained VLMs show limited ability in medical image detection tasks. With SFT, VLMs achieved dramatic improvement in the detection of both healthy anatomical structures and tumors/lesions. However, small object and tumors/lesions detection is still challenging.

### 3.2 T/L Size Estimation Performance

**Baselines vs SFT Models:** Baseline models struggle with T/L size estimation tasks, with MRE ranging from 50.1% to 116.9% (Table 3). After SFT with **5K** samples, Qwen2.5-VL (7B, 32B) can predict T/L size within 10% MRE for $\sim 20\%$ of samples (MER$_{<0.1}$), with MRE at $\sim 30\%$. The best model (Qwen2.5-VL 32B, SFT) can predict T/L size with an average distance error of 12.8 mm. Figure 6 illustrates the improvement of SFT models for each T/L label.

**OOD Performance:** SFT models also show partial generalizability when evaluated on unseen datasets (Figure 7). While accuracy declines compared to in-distribution settings, the relative improvement over baselines remains evident, implying that learned quantitative reasoning skills transfer across domains to some extent. However, performance drops highlight that dataset diversity and annotation consistency remain critical for robust generalization.

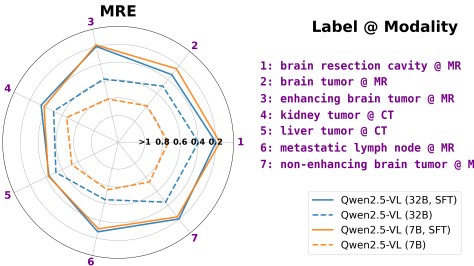

Figure 6: SFT improves T/L size estimation performance.

Table 3: VLM performance on tumor/lesion size estimation tasks. Mean relative error (MRE), success rate (SR), and $MRE_{<k}$ are reported in %, while mean absolute error (MAE) is in millimeters. Improvement after SFT is shown in parentheses.

| Model | MAE ↓ | MRE ↓ | SR ↑ | $MRE_{<0.1}$ ↑ | $MRE_{<0.2}$ ↑ | $MRE_{<0.3}$ ↑ |
|---|---|---|---|---|---|---|
| Qwen2.5-VL (7B) | 32.3 | 71.5 | 100.0 | 0.3 | 1.7 | 3.2 |
| Qwen2.5-VL (32B) | 24.2 | 51.9 | 100.0 | 2.7 | 11.8 | 19.9 |
| Lingshu (32B) | 23.9 | 66.1 | 100.0 | 10.4 | 29.7 | 43.4 |
| InternVL3 (38B) | 22.6 | 50.1 | 100.0 | 4.5 | 15.0 | 24.2 |
| Gemma3 (27B) | 30.7 | 70.7 | 100.0 | 1.4 | 4.9 | 8.7 |
| MedGemma (4B) | 38.9 | 116.8 | 100.0 | 1.1 | 4.5 | 7.6 |
| MedGemma (27B) | 38.9 | 116.9 | 100.0 | 1.1 | 4.3 | 7.5 |
| Llama3.2-Vision (11B) | 25.7 | 61.9 | 99.1 | 4.6 | 14.0 | 21.1 |
| LLava-OneVision (72B) | 26.2 | 83.0 | 100.0 | 4.8 | 17.7 | 29.4 |
| LLaVA-Med-v1.5 (7B) | 48.8 | 74.7 | 22.6 | 0.3 | 0.6 | 1.1 |
| MedDr (40B) | 30.1 | 73.8 | 100.0 | 1.6 | 4.7 | 9.4 |
| HuatuoGPT-Vision (34B) | 28.9 | 89.1 | 100.0 | 8.5 | 26.2 | 42.3 |
| HealthGPT-L14 | 23.6 | 61.3 | 98.9 | 8.0 | 22.6 | 35.8 |
| Qwen2.5-VL (7B, $SFT_{5K}$) | 13.2 (-19.1) | 30.6 (-40.9) | 100.0 | 20.8 (+20.5) | 41.5 (+39.8) | 62.1 (+58.9) |
| Qwen2.5-VL (32B, $SFT_{5K}$) | 12.8 (-11.4) | 30.2 (-21.7) | 100.0 | 19.6 (+16.9) | 43.2 (+31.4) | 63.2 (+43.3) |

**Conclusion:** Although pretrained VLMs can follow instructions to output numerical values in required formats, they fail to estimate T/L size accurately. With SFT, VLMs can reduce MRE from $> 50\%$ to $\sim 30\%$ and show certain generalizability on OOD data.

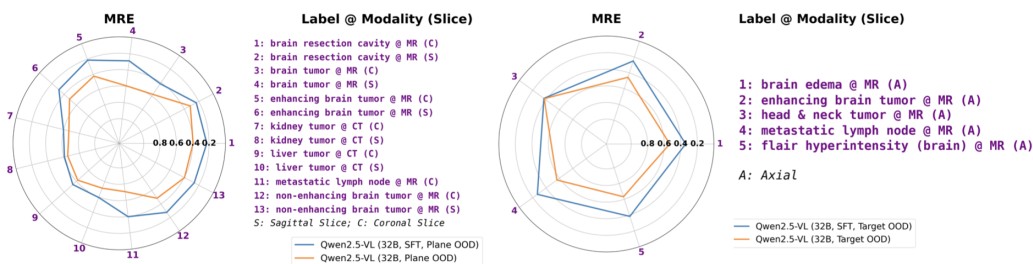

Figure 7: T/L size estimation performance on plane-OOD (left) and target-OOD (right) data.

### 3.3 A/D MEASUREMENT PERFORMANCE

**Baselines vs SFT Models:** Baselines perform poorly on A/D measurement tasks, with distance error ranging from 17.3 mm to 81.6 mm and angle error from 29.9° to 66.1° (Table 4). After SFT with **5K** samples, the distance error reduces to less than 4.5 mm and angle error to less than 4°. Significant improvement in MRE is shown in Figure 8. The distributions of ground truths and VLM predictions are more closely aligned in the SFT models. (Appendix A.7).

**Failure Model Analysis:** Baselines perform poorly on A/D measurement tasks, often predicting only a limited set of values (Appendix A.7). As shown in Figure 8, VLMs fail on certain angle

Table 4: VLM performance on angle and distance measurement tasks. Relative errors (RE), $RE_{<0.1}$, and success rate (SR) are reported in %. Absolute errors (AE) are given in millimeters (distance) and degrees (angle).

| Model | Ceph-Bio-400 | | | | | | | | FeTA24 | | | |
| --- | --- | --- | --- | --- | --- | --- | --- | --- | --- | --- | --- | --- |
| | Distance | | | | Angle | | | | Distance | | | |
| | AE ↓ | RE ↓ | SR ↑ | $RE_{<0.1}$ ↑ | AE ↓ | RE ↓ | SR ↑ | $RE_{<0.1}$ ↑ | AE ↓ | RE ↓ | SR ↑ | $RE_{<0.1}$ ↑ |
| Qwen2.5-VL (7B) | 56.3 | 80.1 | 100 | 0.9 | 55.1 | 3691 | 98.7 | 3.4 | 27.5 | 61.6 | 100 | 2.0 |
| Qwen2.5-VL (32B) | 59.7 | 86.0 | 100 | 0.2 | 41.0 | 3287 | 100 | 2.0 | 19.5 | 49.6 | 100 | 6.0 |
| Lingshu (32B) | 17.3 | 26.1 | 100 | 15.5 | 31.2 | 4648 | 100 | 10.4 | 38.5 | 120.9 | 100 | 0.0 |
| InternVL3 (38B) | 22.1 | 38.2 | 100 | 11.1 | 66.1 | 4405 | 100 | 4.3 | 19.7 | 54.7 | 100 | 11.0 |
| Gemma3 (27B) | 22.6 | 32.3 | 100 | 22.4 | 33.6 | 7862 | 100 | 13.8 | 18.1 | 40.2 | 100 | 15.0 |
| MedGemma (4B) | 43.4 | 67.9 | 100 | 6.5 | 30.5 | 2780 | 100 | 5.8 | 26.2 | 60.4 | 100 | 5.0 |
| MedGemma (27B) | 43.9 | 68.8 | 100 | 6.4 | 29.9 | 2168 | 100 | 6.4 | 26.2 | 60.4 | 100 | 5.0 |
| Llama3.2-Vision (11B) | 81.6 | 117.4 | 100 | 2.9 | 55.3 | 9318.6 | 100 | 8.0 | 29.9 | 73.4 | 100 | 2.0 |
| LLava-OneVision (72B) | 36.3 | 67.8 | 100 | 18.8 | 62.6 | 12269.1 | 100 | 0.0 | 46.0 | 130.7 | 100 | 4.0 |
| MedDr (40B) | 33.7 | 53.6 | 100 | 6.0 | 54.8 | 9149.2 | 100 | 0.2 | 37.1 | 78.5 | 100 | 5.0 |
| HuatuoGPT-Vision (34B) | 45.8 | 65.2 | 100 | 2.8 | 65.8 | 5741.1 | 100 | 9.0 | 40.5 | 83.3 | 100 | 15.0 |
| HealthGPT-L14 | 66.8 | 97.5 | 100 | 1.1 | 42.9 | 434.6 | 100 | 0.0 | 32.1 | 79.4 | 100 | 5.0 |
| Qwen2.5-VL (7B, $SFT_{5K}$) | 3.5 | 5.4 | 100 | 86.4 | 3.6 | 126.8 | 100 | 50.1 | 4.1 | 13.1 | 100 | 49.0 |
| Qwen2.5-VL (32B, $SFT_{5K}$) | 4.5 | 7.9 | 100 | 83.5 | 4.0 | 540.2 | 100 | 50.5 | 4.1 | 14.0 | 100 | 54.0 |

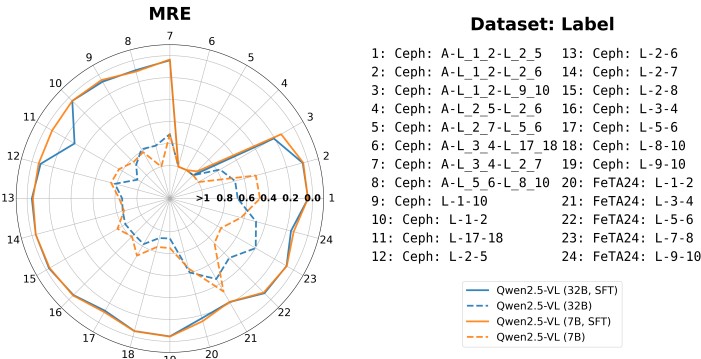

Figure 8: SFT improves VLM performance in A/D size measurement tasks. Labels start with "A" and "L" are angle and distance measurements, respectively. For example, "L-1-2" refers to distance between landmarks 1 and 2.

measurements (labels 4, 5, 6). These correspond to small angles ($< 10°$), which are difficult for models to distinguish and predict accurately.

**Conclusion:** Pretrained VLMs fail to measure angles and distances accurately. SFT can significantly improve performance, reducing relative error to below 10%. Yet, certain small-angle measurements remain challenging.

## 4 CONCLUSION

We introduce MedVision, a large-scale, multi-modality, multi-anatomy medical imaging dataset with rich quantitative annotations for detection and measurement tasks. Using MedVision, we establish a comprehensive benchmark to evaluate VLMs on detection, tumor/lesion size estimation, and angle/distance measurement. Our results show that pretrained VLMs perform poorly on these quantitative tasks, whereas supervised fine-tuning with MedVision substantially improves accuracy. Nonetheless, challenges remain, particularly in detecting and measuring small structures. We expect MedVision to provide a valuable foundation for advancing quantitative capabilities in medical VLMs.

## 5 REPRODUCIBILITY STATEMENT

We will release our entire pipeline of dataset, code, and model checkpoints to facilitate reproducibility. A dataset codebase is provided to reproduce and extend the MedVision dataset.

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

# A APPENDIX

## A.1 DATASET EXAMPLES

Figure 9 shows preprocessed images of each datasets and example of three types of quantitative annotations.

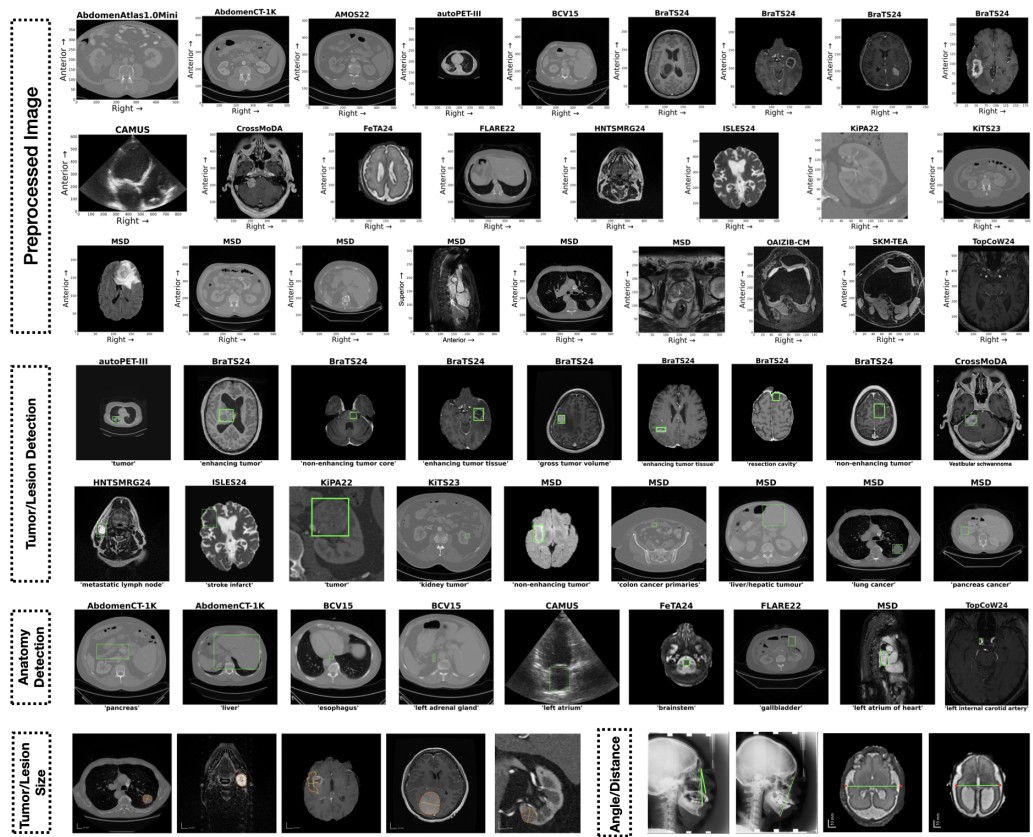

Figure 9: Example preprocessed images and annotations from MedVision dataset.

## A.2 VLMS VERSION

Table 5 lists the HuggingFace model IDs of the evaluated open-weight VLMs.

## A.3 PROMPT TEMPLATES

We designed task-specific prompts to guide the models in generating relevant outputs. Prompts were crafted to be clear and concise, providing necessary context without overwhelming the model.

**Detection Task Prompt.** For detection tasks, the prompt was designed to instruct the model to identify and localize specific anatomical structures or lesions within the image. The model is expected to return the normalized coordinates of the lower-left and upper-right corners of the bounding box. The prompt template is as follows:

```
Task:
Given the input medical image: {image_description},
return the coordinates of the lower-left and upper-right corner
```

Table 5: HuggingFace model ID of the evaluated open-weight VLMs.

| Model | HuggingFace Model ID |
|---|---|
| Qwen2.5-VL (7B) | Qwen/Qwen2.5-VL-7B-Instruct |
| Qwen2.5-VL (32B) | Qwen/Qwen2.5-VL-32B-Instruct |
| Lingshu (32B) | lingshu-medical-mllm/Lingshu-32B |
| InternVL3 (38B) | OpenGVLab/InternVL3-38B |
| Gemma3 (27B) | google/gemma-3-27b-it |
| MedGemma (4B) | google/medgemma-4b-it |
| MedGemma (27B) | google/medgemma-27b-it |
| Llama3.2-Vision (11B) | meta-llama/Llama-3.2-11B-Vision-Instruct |
| LLava-OneVision (72B) | llava-hf/llava-onevision-qwen2-72b-ov-hf |
| MedDr (40B) | Sunanhe/MedDr_0401 |
| HuatuoGPT-Vision (34B) | FreedomIntelligence/HuatuoGPT-Vision-34B |
| HealthGPT-L14 | lintw/HealthGPT-L14 |

```
of the bounding box for the {label_name}.
Format requirement:
{format_prompt}.
```

The format prompt for this task is as follows:

```
The answer should be four decimal numbers separated by commas
without any units or additional text.
The first two numbers are the coordinates of the lower-left
corner and the last two numbers are the coordinates of the
upper-right corner of the bounding box.
Use relative coordinates in the image space, where the origin
is at the lower-left corner of the image. Relative coordinates
should be values between 0 and 1, representing the relative
positions in the image.
```

**Tumor/Lesion Size Measurement Prompt.** For tumor/lesion size measurement tasks, the prompt instructs the model to measure the longest and shortest diameters of the specified tumor/lesion. The model should return the lengths in millimeters. The prompt template is as follows:

```
Task:
Given the input medical image: {image_description},
estimate the major and minor axis lengths of the ellipse enclosing
the {label_name}, in {metric_unit}.
Additional information:
{pixel_size_prompt}.
Format requirement:
{format_prompt}.
```

where we provide the pixel size information in {pixel_size_prompt} following the format:

```
The pixel size for this image is {adjusted_pixel_width}
(width) x {adjusted_pixel_height} (height).
```

Note that the pixel size is adjusted according to the model's image processing strategy. The format prompt for this task is as follows:

```
The answer should be two decimal numbers separated by a comma
```

```
without any units or additional text. The first is the major
axis length, and the second is the minor axis length.
```

**Angle/Distance Measurement Prompt.** For angle/distance measurement tasks, the prompt instructs the model to measure specific anatomical angles or distances based on provided landmark names. The model should return the measurement in the specified unit. The prompt template is as follows:

```
Task:
Given the input medical image: {image_description},
{task_prompt}
Additional information:
{pixel_size_prompt}
Format requirement:
{format_prompt}.
```

An example of {task_prompt} for distance measurement is:

```
estimate the distance of {distance_name} in {metric_unit},
which is the distance between 2 landmark points:
(landmark 1) {p1},
(landmark 2) {p2}.
```

An example of {task_prompt} for angle measurement is:

```
estimate the angle of {angle_name} in {metric_unit},
which is the angle between 2 lines:
(line 1) the line connecting {l1p1} and {l1p2},
(line 2) the line connecting {l2p1} and {l2p2}.
```

The format prompt for this task is as follows:

```
The answer should be a single decimal number
without any units or additional text.
```

### A.4 VLM Image Processing Strategy

Different VLMs define their own image processing strategy in the image processor classes (or equivalent modules), where image processing pipelines including resizing, cropping, padding, and rescaling are defined. We summarized the image resizing stategies of the evaluated VLMs in Table 6, upon which we designed appropriate pixel size adjustment for conveying physical spacing information to VLMs.

### A.5 Details of Detection Performance

#### A.5.1 Anatomy Level Detection Performance

Figure 10 shows the detection performance of VLMs in various anatomical groups. Labels are first categorized into these anatomy groups. Then, weighted average metrics are calculated for each group. Tumor/lesion labels are highlighted in color.

A substantial improvement is observed in the SFT models across all metrics. In Figure 10, anatomical groups are arranged in descending order of recall. Overall, detection performance is lower for tumor/lesion groups than for healthy anatomy, likely due to greater variability in tumor size, shape, and location. The limited number of tumor/lesion samples may also contribute to reduced performance.

Table 6: Image resize strategies of VLMs. ‡ aspect ratio preserved

| Model | Image Size | Image Resize Strategy |
|---|---|---|
| Qwen2.5-VL | dynamic | reshape to a size divisible by 28x28 patch |
| Lingshu | dynamic | reshape to a size divisible by 28x28 patch |
| InternVL3 | 448x448 | reshape to a fixed size |
| Gemma3 | 896x896 | reshape to a fixed size |
| MedGemma | 896x896 | reshape to a fixed size |
| Llama3.2-Vision | dynamic | reshape‡ to fit in a tiled canvas (with patches of size 560x560) |
| LLava-OneVision | dynamic | reshape‡ to fit in a tiled canvas (with patches of size 384x384) |
| LLaVA-Med | 336x336 | reshape to a fixed size |
| MedDr | 448x448 | reshape to a fixed size |
| HuatuoGPT-Vision | 336x336 | reshape to a fixed size |
| HealthGPT-L14 | 336x336 | reshape to a fixed size |

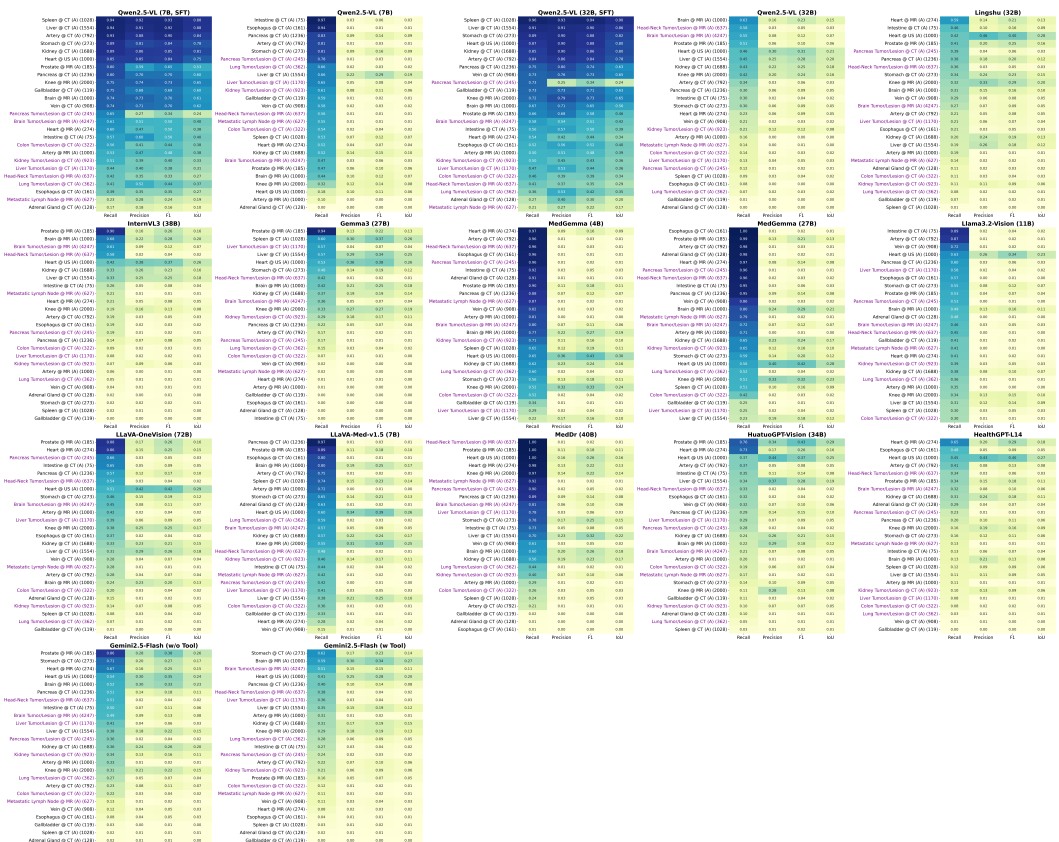

Figure 10: Detection performance of VLMs at anatomy level. Tumor/lesion labels are highlighted in color.

### A.5.2   LABEL LEVEL DETECTION PERFORMANCE

Figure 11 shows the label level detection performance of VLMs at each box-image-ratio group. Box-image-ratio is defined as the area ratio of the bounding box to the image. Labels are grouped into different box-image-ratio ranges. We visualized the composition of box-image-ratio groups for each label at the bottom of Figure 11. For each box-image-ratio group and label combination, we visualized recall, precision, and F1 score. Figure 11 is a detailed map of VLM performance at various target sizes and labels. We observed that detection performance generally improves with larger box-image-ratio. Significant performance improvement is observed in the SFT models.

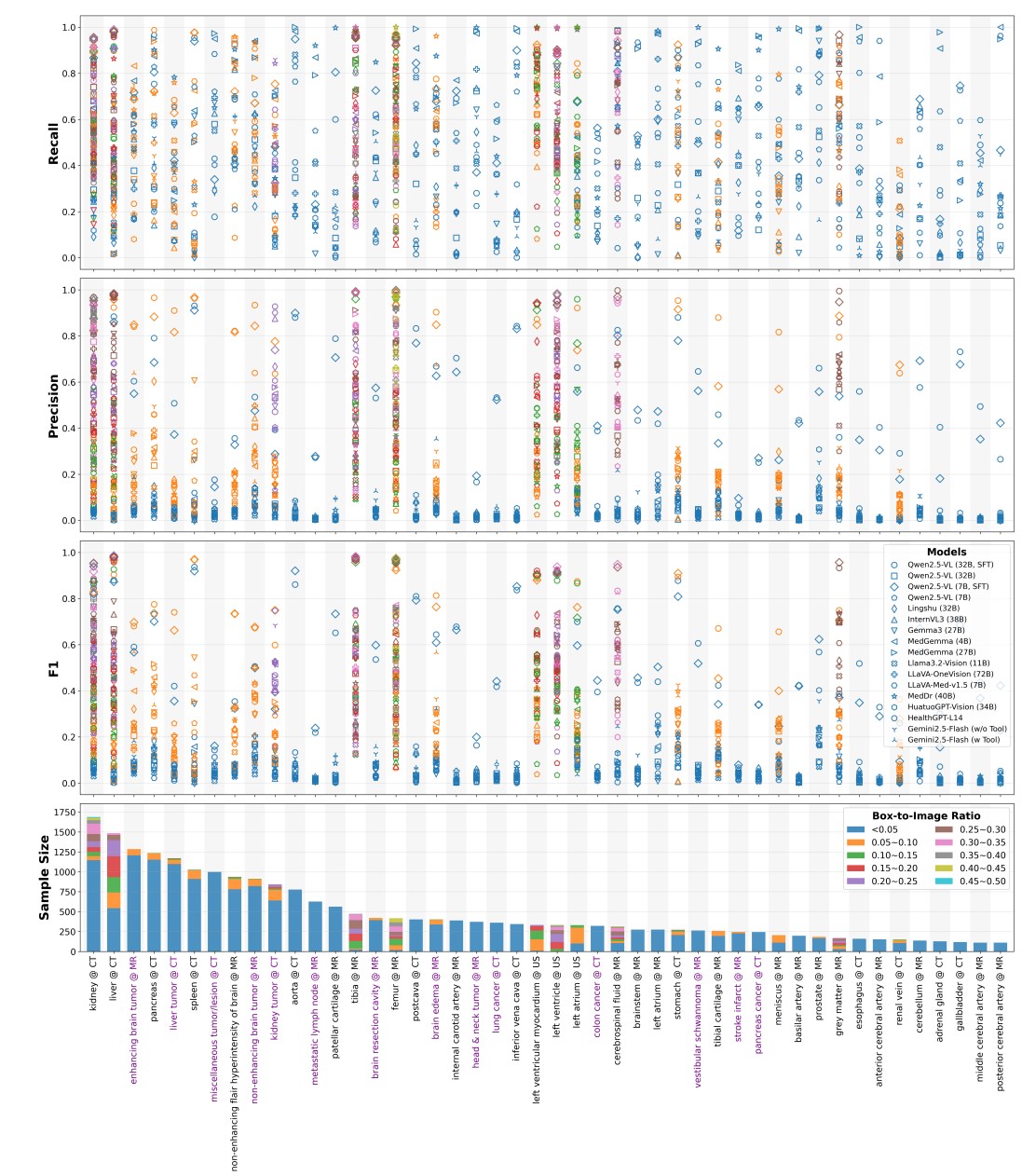

Figure 11: Label level detection performance at each box-image-ratio group.

### A.5.3 EFFECT OF TARGET SIZE

Figure 11 shows a trend of increasing detection performance with larger target sizes (box-image-ratio). To further illustrate this effect, we grouped all detection targets into different box-image-ratio groups, and calculated the overall detection performance for each group. As shown in Figure 12, precision and recall is positively correlated to relative target size (box-image-ratio). The SFT models consistently outperform the baselines across all target sizes.

**VLMs vs Random Guess Model:** We compared the detection performance of baselines and SFT VLMs with a random guess model. Most pretrained models perform slightly better than random guess, while the medical VLM HuatuoGPT-Visoin (34B) standouts from the rest baselines.

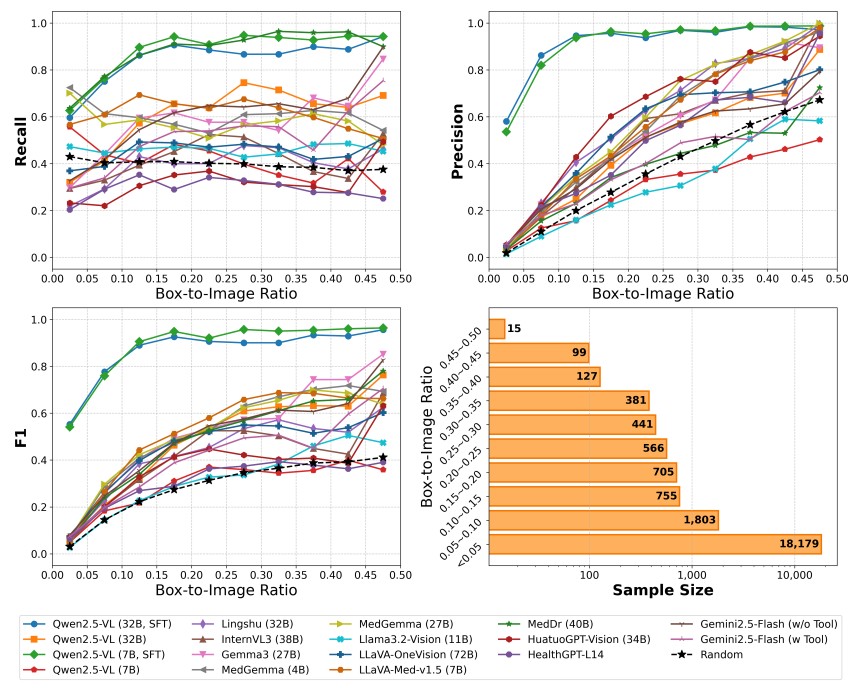

Figure 12: Effect of target size, quantified by box-image-ratio, on detection performance of VLMs.

## A.6 DETAILS OF T/L SIZE ESTIMATION PERFORMANCE

Figure 13 shows the distribution of ground truths and VLM predictions in tumor/lesion size estimation tasks.

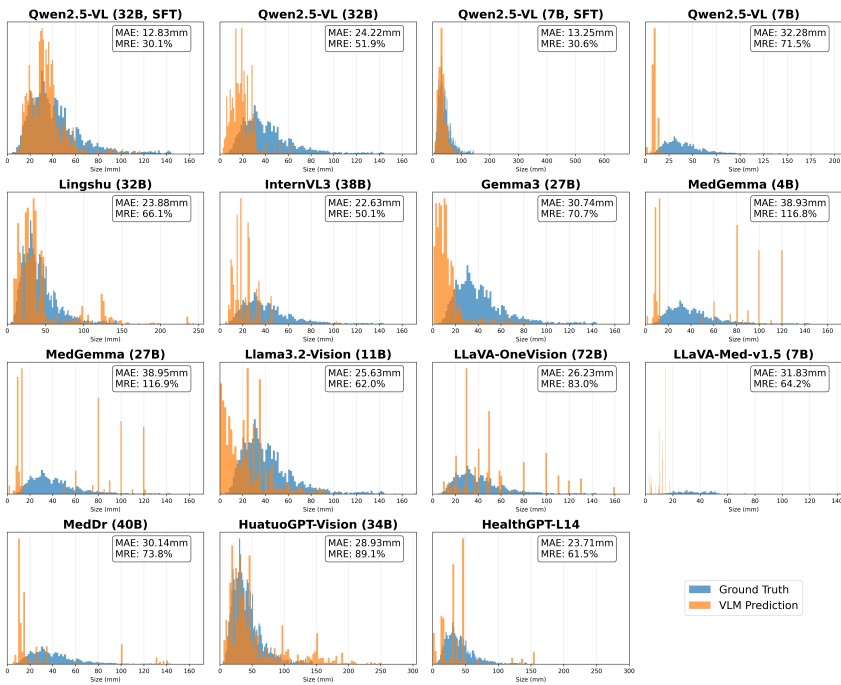

Figure 13: Distribution of ground truths and VLM predications in tumor/lesion size estimation tasks.

### A.7 DETAILS OF A/D MEASUREMENT PERFORMANCE

Figure 14 and Figure 15 show the distribution of ground truths and VLM predictions in angle and distance measurement tasks, respectively. We observed a common behavior among baselines where VLMs tend to predict a limited set of values, shown as spikes in the distribution plots. With SFT, the diversity of model predictions increases and better aligns with the ground truth distribution.

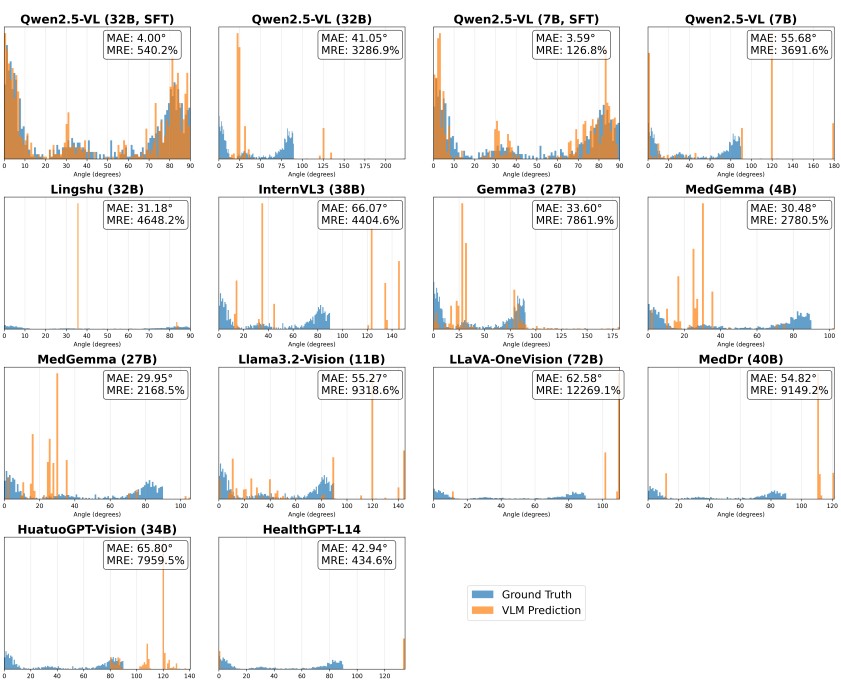

Figure 14: Distribution of ground truths and VLM predictions in angle measurement tasks.

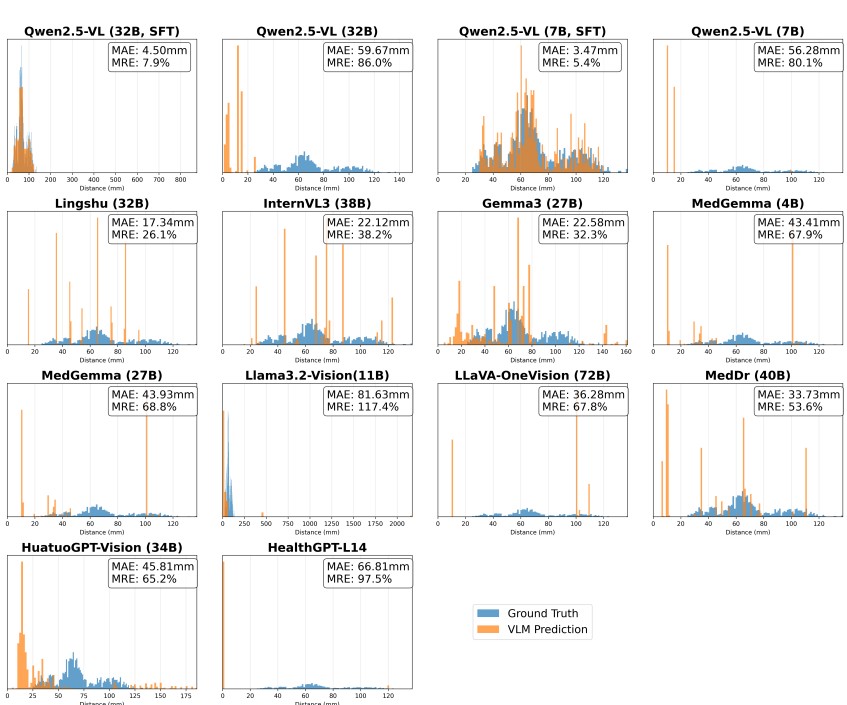

Figure 15: Distribution of ground truths and VLM predictions in distance measurement tasks.

