# OpenReview forum: "MedVision: Dataset and Benchmark for Quantitative Medical Image Analysis"
_ICLR.cc/2026/Conference — ICLR 2026 Conference Withdrawn Submission_

### Official Review · Reviewer_mhtK · 2025-10-29

**Soundness:** 2
**Presentation:** 3
**Contribution:** 4
**Rating:** 2
**Confidence:** 4

**Summary:**

This paper focuses on quantitative medical image analysis and introduces the MedVision dataset/benchmark, which defines three representative tasks: (1) detection of anatomical structures and abnormalities, (2) tumor/lesion (T/L) size estimation, and (3) angle/distance (A/D) measurement. The authors evaluate existing vision-language models (VLMs) in both zero-shot and supervised fine-tuning (SFT) settings, demonstrating that SFT can effectively improve model performance on these quantitative tasks.

**Strengths:**

1. The research motivation is well aligned with practical clinical needs, and the designed tasks are highly valuable and relevant to real-world medical applications.
2. The collection of data containing physical measurements to construct the dataset is a reasonable and meaningful approach.

**Weaknesses:**

1. It is unclear whether quantitative assessment is an intrinsic capability that VLMs should possess. Would it be more practical to perform these tasks using a segmentation model combined with rule-based computation, especially since the current dataset is constructed from segmentation data? Perhaps VLMs would be better suited to handling such tasks in an agent-based paradigm.
2. Open-ended VQA may not be the most appropriate evaluation setting, as quantitative reasoning requires structured information extraction. It is recommended to include closed-ended VQA tasks so that both open- and closed-ended settings jointly form a more comprehensive benchmark.
3. Line 155: Focusing solely on single instance detection is unreasonable, especially for abnormality detection. It is suggested that multi-instance scenarios be incorporated into the dataset and benchmark.
4. Line 174: For the A/D annotation task, it would be helpful to provide a table explaining the task motivation, e.g., which specific angles or distances are clinically relevant for diagnosing certain diseases.
5. Line 187: The random split raises concerns about its rationality. If the collected public datasets already provide official splits, why not use them?
6. Line 265: For the evaluation metrics, more detailed calculation formulas should be provided to ensure clarity and reproducibility.
7. Line 257: Normalizing coordinates to the range [0, 1] is problematic. Most modern VLMs operate with integer-normalized coordinates in [0, 1000], as they handle small decimal values poorly (e.g., confusing “0.11 > 0.9”). Using [0, 1] normalization could lead to unreliable detection results (as seen in Table 2).

**Questions:**

Same as those raised in the Weaknesses section.

---

### Official Review · Reviewer_bYCv · 2025-10-29

**Soundness:** 3
**Presentation:** 3
**Contribution:** 3
**Rating:** 6
**Confidence:** 4

**Summary:**

This paper proposes MedVision, a large-scale multimodal medical QA benchmark. The benchmark is repurposed from other 22 open-sourced related datasets. Different from other benchmarks, MedVision focuses not only on detection of anatomical structures
and abnormalities but also the estimation of tumor size as well as angle/distance measurement. Evaluation using a wide range of open-sourced/frontier models demonstrates the difficulty of the benchmark and the lack of medical reasoning capabilities. Furthur experiments show that SFT signficiantly improves performance.

**Strengths:**

1) The paper addresses an important area (aka medical image analysis) that could facilitate the development of large medical VLMs and drive better clinical decision-making. Although the data is not new, the authors did manually re-label many different image pairs.
2) The quantitative evaluation of size and angle is new and important. The results also demonstrate that current models struggle on these tasks.
3) Flow of the paper is good. Easy to read and understand.

**Weaknesses:**

1) Failure analysis needs improvement. The authors should isolate the effect of geometric reasoning from visual perception. The authors discuss failure modes like small-object detection and angle/value collapse, but it is still unclear if the model simply fails at perception or fails at reasoning (unit handling, geometry).

***One experiment you can try is to (on a small set of course) feed GT BBbox to the model to re-measure size / angle / distance, this would give a clear picture of exactly where the model fails.

2) Following 1), there is also the regular "reasoning" (natural language reasoning). In the current evaluation, it seems like the prompt asks the model to directly output the answer without explanation. I wonder if the reasoning process help (although unlikely, but it'd be interesting to see).

***This could be simply done by prompting the model to reason first then answer.

3) Tool-using results are very surprising. In Table 2, the Gemini model with tool-using performance is worse compared to the oracle model. This is very surprising as tool-using, such as writing code for better cropping, etc should largely benefit these kind of tasks at least in my own experience. I don't think this is a weakness per se if the authors can clearly explain why tool-using doesn't help and even lowers acc; this would also immensely elevate the impact of this work.

***The authors could show a few example to help explain

**Questions:**

1) I think it'd be good to include a GPT baseline, which will make the results more comparative and intuitive to field workers.
2) Recent works have shown that RLVR improves medical reasoning [1][2]/generic visual grounding tasks much more than SFT, while also producing readable reasoning processes. Thus I believe It'd make the paper more comprehensive to include an RFT result, if their compute allows.


[1] Pan et. al. MedVLM-R1: Incentivizing Medical Reasoning Capability of Vision-Language Models (VLMs) via Reinforcement Learning
[2] Lai et. al. Med-R1: Reinforcement Learning for Generalizable Medical Reasoning in Vision-Language Models

**Details Of Ethics Concerns:**

There is no ethics or legal statement in this paper as far as I checked. If the authors only corrected data from publicly available datasets, then it might not be much of an issue, but it is safe to check. The authors are also strongly recommended to include an ethics statement given the nature of this work.

---

### Official Review · Reviewer_JyAs · 2025-10-30

**Soundness:** 2
**Presentation:** 2
**Contribution:** 2
**Rating:** 2
**Confidence:** 4

**Summary:**

The paper presents MedVision, a large-scale dataset and benchmark aimed at evaluating and enhancing vision-language models (VLMs) for quantitative medical image analysis. It includes 22 public datasets with 30.8 million image-annotation pairs across diverse anatomies and modalities. The study focuses on three quantitative tasks, including detection, tumor/lesion size estimation, and angle/distance measurement and shows that existing VLMs perform poorly on these tasks. With supervised fine-tuning on MedVision, the models achieve notable improvements in quantitative accuracy and precision.

**Strengths:**

The paper is easy to follow, with well-structured descriptions of datasets and experiments. The benchmark design is systematic, providing a valuable foundation for evaluating quantitative reasoning in medical VLMs.

**Weaknesses:**

However, there exist the following concerns.
The contribution remains limited, as the work mainly consolidates existing datasets and automatically extracts bounding boxes and bidirectional tumor/lesion sizes. The dataset construction process is more of an engineering integration rather than a methodological or conceptual advancement.
The question design in Section 3 centers on dataset annotation rather than clinical reasoning. It does not clearly connect to diagnostic decision-making or show whether the proposed framework meaningfully improves clinical outcomes or supports physicians’ decision processes.
Many of the proposed quantitative tasks, such as size or distance measurement, could also be achieved through standard segmentation models or smaller specialized networks. The paper does not clarify what advantages large-scale VLMs bring beyond these existing methods.
The dataset lacks richer contextual information like radiology reports, diagnostic impressions, or lesion-level explanations. Compared with previous benchmark papers, it provides limited new insights into clinical reasoning or multimodal understanding beyond simple quantitative annotation.

**Questions:**

The question design in Section 3 centers on dataset annotation rather than clinical reasoning. It does not clearly connect to diagnostic decision-making or show whether the proposed framework meaningfully improves clinical outcomes or supports physicians’ decision processes.
Many of the proposed quantitative tasks, such as size or distance measurement, could also be achieved through standard segmentation models or smaller specialized networks. The paper does not clarify what advantages large-scale VLMs bring beyond these existing methods.
The dataset lacks richer contextual information like radiology reports, diagnostic impressions, or lesion-level explanations. Compared with previous benchmark papers, it provides limited new insights into clinical reasoning or multimodal understanding beyond simple quantitative annotation.

---

### Official Review · Reviewer_5WgS · 2025-11-03

**Soundness:** 3
**Presentation:** 3
**Contribution:** 2
**Rating:** 4
**Confidence:** 4

**Summary:**

This paper introduces MEDVISION, a large composite dataset that integrates 22 publicly available datasets, along with a benchmark for the quantitative evaluation of medical vision-language models (VLMs). The authors curated the original annotations from each dataset and focused on three key  tasks: (1) detection of anatomical structures and abnormalities, (2) tumor/lesion (T/L) size estimation, and (3) angle/distance (A/D) measurement. I appreciate the authors’ efforts in evaluating a wide range of VLMs and assembling such a comprehensive dataset. However, I have several concerns regarding this work, primarily related to the composition of the dataset and the formulation

**Strengths:**

MEDVISION integrates a diverse collection of publicly available datasets, which could serve as a valuable resource for the medical vision-language community.
The authors conducted extensive experiments by evaluating a wide range of VLM models, providing useful baselines for future research.

**Weaknesses:**

While MEDVISION aggregates a large number of datasets, the curated annotations do not represent a significant improvement over the original data — in some cases, they are even of lower quality. For instance, the annotation process excluded 2D slices containing multiple bounding boxes, which may limit the dataset's applicability and realism. The authors justify this by stating that the study focuses on single-instance detection; however, this raises the question of clinical relevance — what is the practical value of single-instance detection in real-world clinical settings?

This concern extends to the overall task formulation. It is unclear why large VLMs are needed to perform tasks such as detection, size estimation, and angle/distance measurement, which are traditionally and effectively addressed using standard detection, segmentation, and keypoint detection models. The paper would benefit from a stronger justification for the use of VLMs in this context, as well as a clearer articulation of how these tasks benefit from vision-language modeling.

**Questions:**

1. It appears that the curated annotations do not significantly improve upon the original data — in some cases, they seem to be simplified derivations (e.g., generating bounding boxes from segmentation masks). Could the authors clarify what value this curation adds, especially when it may result in a loss of information or clinical realism?

2. Why are VLMs needed for these tasks, given that standard detection, segmentation, and keypoint detection models can already handle them effectively? What unique advantages do VLMs bring to the table in this context?

---

### Note · Authors · 2025-11-13

**Comment:**

We thank the reviewers and the area chair for their time and constructive feedback. We will refine and strengthen the paper based on their comments.

**Withdrawal Confirmation:**

I have read and agree with the venue's withdrawal policy on behalf of myself and my co-authors.